# Synergistic Effect of TiO_2_-Nanoparticles and Plant Growth-Promoting Microorganisms on the Physiological Parameters and Antioxidant Responses of *Capsicum annum* Cultivars

**DOI:** 10.3390/antiox14060707

**Published:** 2025-06-10

**Authors:** Atiya Bhatti, Araceli Sanchez-Martinez, Gildardo Sanchez-Ante, Daniel A. Jacobo-Velázquez, Joaquín Alejandro Qui-Zapata, Soheil S. Mahmoud, Ghulam Mustafa Channa, Luis Marcelo Lozano, Jorge L. Mejía-Méndez, Edgar R. López-Mena, Diego E. Navarro-López

**Affiliations:** 1Tecnologico de Monterrey, Escuela de Ingeniería y Ciencias, Ave. General Ramon Corona 2514, Zapopan 45138, Jalisco, Mexico; atiya.bhatti10@gmail.com (A.B.); gildardo.sanchez@tec.mx (G.S.-A.); gmchanna139@gmail.com (G.M.C.); marcelo.lozano@tec.mx (L.M.L.); 2Departamento de Ingeniería de Proyectos, CUCEI, Universidad de Guadalajara, Av. José Guadalupe Zuno # 48, Industrial los Belenes, Zapopan 45157, Jalisco, Mexico; araceli.sanchez46@academicos.udg.mx; 3Tecnologico de Monterrey, Institute for Obesity Research, Ave. General Ramon Corona 2514, Zapopan 45201, Jalisco, Mexico; djacobov@tec.mx; 4Biotecnología Vegetal, Centro de Investigación y Asistencia en Tecnología y Diseño del Estado de Jalisco A.C., Camino Arenero 1227, El Bajío, Zapopan 45019, Jalisco, Mexico; jqui@ciatej.mx; 5Department of Biology, The University of British Columbia, Okanagan Campus, 1177 Research Road, Kelowna, BC V1V 1V7, Canada; soheil.mahmoud@ubc.ca; 6Programa de Edafología, Colegio de Postgraduados, Campus Montecillo, Carretera México Texcoco km 36.4, Montecillo 56264, Mexico

**Keywords:** biocompatibility, total phenolic compounds, titanium dioxide, antioxidant enzymes, physiological response

## Abstract

Titanium dioxide nanoparticles (TiO_2_-NPs) were synthesized using the molten salt method and systematically characterized. TiO_2_-NPs were evaluated for their capacity to promote the growth of *Capsicum annuum* cultivars together with the plant growth-promoting microorganisms (PGPMs) *Bacillus thuringiensis* (Bt) and *Trichoderma harzianum* (Th). The variables analyzed included physiological parameters and antioxidant responses. The capacity of TiO_2_-NPs to scavenge free radicals was also investigated, along with their biocompatibility, using *Artemia salina* as an in vivo model. The results demonstrated that TiO_2_-NPs exhibited a nanocuboid-type morphology, negative surface charge, and small surface area. It was noted that TiO_2_-NPs enhanced the CFU and spore production of Bt (1.56–2.92 × 10^8^ CFU/mL) and Th (2.50–3.90 × 10^8^ spores/mL), respectively. It was observed that TiO_2_-NPs could scavenge DPPH, ABTS, and H_2_O_2_ radicals (IC_50_ 48.66–109.94 μg/mL), while not compromising the viability of *A. salina* at 50–300 μg/mL. TiO_2_-NPs were determined to enhance the root length and fresh and dry weights of chili peppers. Similarly, TiO_2_-NPs in synergy with Bt and Th increased the activity of β-1,3-Glucanase (2.45 nkat/g FW) and peroxidase (69.90 UA/g FW) enzyme activity, and increased the TPC (29.50 GA/g FW). The synergy of TiO_2_-NPs with the PGPMs consortium also upregulated the total chlorophyll content: 210.8 ± 11.4 mg/mg FW. The evidence from this study unveils the beneficial application of TiO_2_-NPs with Bt and Th as an efficient approach to promote the physiology and antioxidant responses of chili peppers.

## 1. Introduction

Precision agriculture focuses on optimizing crop nutrition, health, and yield through sustainable and efficient approaches [1]. Currently, two major approaches in precision agriculture are plant growth-promoting microorganisms (PGPMs) and nanofertilizers. The former includes a diverse category of bacteria and fungi that can solubilize essential nutrients, enhance the production of hormones during crop growth and vigor, suppress phytopathogens, and upregulate crop resilience to abiotic stresses [2]. Representative beneficial bacteria and fungi employed for these purposes are strains from the genera *Bacillus* (e.g., *B. subtilis*, *B. amyloliquefaciens*, and *B. thuringiensis*) [3], and *Trichoderma* (e.g., *T. harzianum*, *T. viride*, and *T. reesei*) [4]. The latter encompasses nanostructured materials, such as metals, metal oxides, or polymers, to improve homeostatic processes, hormonal balance, and nutrient assimilation in crops. Representative nanomaterials used in precision agriculture include zinc oxide nanoparticles (NPs) [5], lignin nanocapsules [6], air nanobubbles [7], and fluorescent nanosensors [8].

In contrast to other non-metallic compounds, titanium oxide (TiO_2_) possesses several unique physicochemical properties that allow it to be used in various agricultural applications [9,10]. For instance, when exposed to ultraviolet (UV) light, it can act as a photocatalyst and generate reactive oxygen species (ROS), which can decompose organic compounds and pollutants [11]. Similarly, TiO_2_-based nanostructures are preferred because of their high refractive index (2.8 at 500 nm), which helps enhance light absorption efficiency in plants and hence upregulates photosynthesis [12]. TiO_2_-derived nanomaterials also serve as ultraviolet barriers, preventing potential tissue damage and ensuring healthier and more resilient plants [13]. TiO_2_-NPs have been shown to influence gene expression in metabolic pathways associated with photosynthesis, water content balance, and growth promotion of *Solanum lycopersicum* L. [14], *Ocimum tenuiflorum* L. [15], *Medicago sativa* L. [16], and *Triticum aestivum* L. [17]. Despite the wide evidence of TiO_2_-NPs as nanofertilizers, a limited number of studies have partially considered their synergy with PGPMs (i.e., *B. pumilus* and *T. asperellum*) [18,19], and evaluated their effects on other crops of worldwide economic importance, for example, *Capsicum annuum*.

*C. annuum*, frequently known as chili peppers, is a widely used culinary plant from the *Capsicum* genus. Considering the growing conditions, chili peppers are cultivated in regions with warm seasons and well-drained and fertile soils. Representative countries that cultivate and commercialize chili peppers include Mexico, India, and China [20]. In Mexico, the main designations for chili peppers are habanero chili from the Yucatan Peninsula and Yahualica chili from the Los Altos area of Jalisco. In the same country, chili cultivation represents 20.2% of vegetable production nationwide, with an inventory of 64 types of *C. chili* peppers distributed in Oaxaca, Guerrero, Puebla, and Veracruz. In addition to their cultural importance and consumer demand, chili peppers have been widely investigated because of their anti-inflammatory and antioxidant properties, which arise from their capsaicinoids, flavonoids, phenols, and vitamin content [21,22]. Current approaches to promoting the growth of chili peppers include chemical fertilizer application, genetic approaches, and irrigation techniques. Although some have been demonstrated to benefit chili pepper growth and yield, their use can be associated with degradation of soil health, eutrophication, health risks, loss of biodiversity, high operational costs, and labor-intensive practices.

Given the need to develop novel, efficient, and functional alternatives to promote the growth of chili peppers to meet nutritional and food demands, this study investigated the synthesis of TiO_2_-NPs and their effect on physiological parameters and antioxidant responses in chili pepper cultivars together with *B. thuringiensis* (Bt) and *T. harzianum* (Th). The compatibility of TiO_2_-NPs was also tested against *B. subtilis* (Bs). The experiments were conducted in a greenhouse under controlled temperature and humidity. The effect of TiO_2_-NPs and Bt and Th was evaluated in terms of plant height, root length, number of leaves, fresh weight (FW), and dry weight (DW). The antioxidant response of cultivars towards TiO_2_-NPs and Bt and Th was recorded based on variabilities in β-1,3-glucanase (β-1,3-G) and peroxidase (POX) activity. Changes in the total phenolic content (TPC) were also evaluated. The scavenging activity of TiO_2_-NPs was studied using the DPPH, ABTS, and H_2_O_2_ assays. The data retrieved from this study expands the knowledge about nano-fertilizers and their integration with PGPMs for efficient precision agriculture practices in crops of worldwide economic importance.

## 2. Materials and Methods

### 2.1. Material Synthesis

TiO_2_-NPs were prepared using the molten salt method. Briefly, a homogeneous mixture was prepared by mixing and grinding 1 g of titanium (IV) oxide (P25, 99.5%, Sigma-Aldrich, St. Louis, MO, USA) with 4 g of a eutectic mixture (NaCl and KCl, 1:1 M). The mixture was then placed in a crucible dish and calcined for 6 h at 900 °C in air. The solution was washed several times with deionized water to remove any remaining chloride residue. After washing, the powder was dried in air at 100 °C for 6 h. 

### 2.2. Material Characterization

The crystal structures of the TiO_2_-NPs were characterized by X-ray diffraction (XRD) using an Empyrean diffractometer (PANalytical, Cincinnati, OH, USA) equipped with a copper anode (l = 1.5406 Å). XRD patterns were obtained for 2θ ranging from 10° to 90° with a step size of 0.01°. Attenuated total reflection Fourier transform infrared (ATR-FTIR) spectroscopy was used to assess the presence of organic matter in the NP structure. ATR-FTIR spectra were recorded in the 4000–400 cm^−1^ range using an IR Affinity-1S (Shimadzu, Santa Clara, CA, USA) spectrometer. The optical properties were assessed using absorption spectra obtained with a Cary-5000 UV–Vis (Agilent Technologies, Santa Clara, CA, USA) spectrometer equipped with a polytetrafluoroethylene integration sphere. Absorbance spectra were recorded in the range of 200–800 nm. The particle size and zeta potential of the liquid suspension (1 mg/mL) were measured at 25 °C using a Zetasizer Pro instrument (Malvern Instruments, Almelo, The Netherlands). The morphology of the NPs was investigated using field-emission scanning electron microscopy (FESEM) (TESCAN MIRA3 model, Ciudad de México, Mexico). The Brunauer–Emmett–Teller (BET) method was used to determine the specific surface area (SBET) using a Bel-Japan MiniSorp II instrument (Osaka, Japan).

### 2.3. Culture of PGPMs and Compatibility Assays

#### 2.3.1. Culture of PGPMs

The PGPMs used in this study were Bs (ATCC 6633), Bt (B-BT0001), and Th (F-BT0002), which were isolated from soil and deposited in the Tecnologico de Monterrey ceparium. Briefly, Bt was inoculated in LB broth and incubated overnight at 30 °C with shaking at 180 rpm. Th was inoculated in Petri dishes containing PDA and incubated for six days at 30 °C.

#### 2.3.2. Compatibility Assay Between TiO_2_-NPs and Bs and Bt

The compatibility of TiO_2_-NPs with Bs and Bt was investigated using a kinetic assay and the spread plate method. The former was performed by placing Bt and Bs in a 96-well microplate together with 50, 100, and 150 μg/mL TiO_2_-NPs at a final volume of 200 μL LB. Growth was monitored and measured using a microplate reader at 600 nm every hour for 24 h. The latter was executed based on the effect of TiO_2_-NPs on colony-forming units (CFUs), which were considered in previous reports, with slight modifications [23]. The experiment was conducted in Petri dishes containing two LB agar layers: the bottom layer consisted of only LB agar and the top layer contained 50, 100, and 150 μg/mL TiO_2_-NPs. Next, 100 μL of a 10^−6^ dilution of the overnight-grown culture of Bt or Bs was spotted on the surface of the agar and spread with a bent glass rod. Colonies were counted after 24 h of incubation at 30 °C, and the total CFU/mL was calculated using Equation (1).(1)$$\frac{CFU}{mL}=\frac{\left(Number\;of\;colonies\right)\left(Dilution\;factor\right)}{Volume\;of\;culture\;plated}$$

Determination of Bt CFU after treatment with TiO_2_-NPs.

#### 2.3.3. Compatibility Assay Between TiO_2_-NPs and Th

The compatibility effect of TiO_2_-NPs with Th was investigated by placing a square of 0.5 cm of Th in PDA supplemented with 50, 100, and 150 μg/L of TiO_2_-NPs, respectively. The plates were incubated for six days at 30 °C until mycelium growth and spore production were observed. Then, 5 mL of sterile distilled water was added to the agar surface, and the spores were scraped off gently with a brush to ensure that the mycelium remained undisturbed. The resulting spore suspension was then collected in a 50 mL Falcon tube. Fungal spores were counted using a hemocytometer (Neubauer, Grid Optik), according to standard procedures [24]. The results obtained from this experiment were reported as spores per mL using Equation (2). For the experiment with Bt, ampicillin was used as the positive control, whereas terbinafine was used for the compatibility assay against Th.(2)$$\frac{Spores}{mL}=\frac{\left(Number\;of\;spores\right)\;\left(10^{4}\right)\left(Dilution\;factor\right)}{Number\;of\;squares}$$

Determination of Th spore production after treatment with TiO_2_-NPs.

### 2.4. Antioxidant Activity

The antioxidant capacity of TiO_2_-NPs was analyzed using the DPPH, ABTS, and H_2_O_2_ assays. Briefly, the DPPH assay was performed by dissolving 4 mg of DPPH reagent in ethanol, stirring for 2 h, and measuring its absorbance until it reached 0.9. DPPH solution (200 μL of DPPH solution) was then mixed with 20 μL of TiO_2_-NPs at 50, 100, 150, 200, 250, and 300 μg/mL. The mixture was vortexed and stored in the dark for 30 min. The absorbance of the samples was measured at 517 nm using a Cary 60 UV–Vis spectrophotometer (Agilent Technologies, Santa Clara, CA, USA). The ABTS assay was performed by dissolving 19.7 mg of ABTS reagent and 186.2 mg of potassium persulfate. The mixture was stirred overnight for 1 h in the dark. The next day, 200 μL of the resultant solution was mixed with 20 μL of 50, 100, 150, 200, 250, or 300 μg/mL TiO_2_-NPs. The samples were vortexed and maintained in the dark for 6 min, and the absorbance was recorded at 720 nm using an exact spectrophotometer. The H_2_O_2_ assay was conducted by mixing 70 μL of H_2_O_2_ solution (40 mmol/L) with 100 μL of TiO_2_-NPs at 50, 100, 150, 200, 250, and 300 μg/mL concentrations. The samples were preserved in the dark for 30 min, and the absorbance was determined at 230 nm using an exact spectrophotometer. Quercetin (Qu) was used as a positive control for all three assays. The radical scavenging activity of TiO_2_-NPs was calculated using Equation (3), where A_0_ was associated with the absorbance of DPPH, ABTS, or H_2_O_2_ without treatment, and A_1_ was correlated with the absorbance of DPPH, ABTS, or H_2_O_2_ with treatment. All experiments were performed in triplicate.(3)$$\%Scavenging\;Activity=\frac{\left(A_{0}-A_{1}\right)}{A_{0}}\;\times \;100$$

Determination of DPPH, ABTS, and H_2_O_2_ scavenging activities of TiO_2_-NPs.

### 2.5. Evaluation of the Toxicity of TiO_2_-NPs in A. salina Nauplii

The toxicity of TiO_2_-NPs was investigated in *A. salina* nauplii. Briefly, *A. salina* cysts obtained from a commercial supplier (Puebla, Mexico) were dispensed into distilled water supplemented with 35 g of artificial sea salt and preserved under constant illumination for 48 h. Next, 250 μL of nauplii was dispensed (per well) into a 96-well plate and treated with 50, 100, 150, 200, 250, and 300 μg/mL of TiO_2_-NPs. Exposure to the treatment was maintained for 24 h, and possible morphological aberrations were studied using a Leica DMi1 inverted microscope (Leica, Wetzlar, Germany) equipped with a FLEXACAM C1 camera. Potassium dichromate (K_2_Cr_2_O_7_) was used as a positive control. All experiments were performed in triplicate.

### 2.6. Preparation of Plant Material: Physiological Evaluation and Antioxidant Responses Upon Treatment with TiO_2_-NPs and Inoculation with Bt and Th

Serrano chili seeds (cv. Camino Real) were disinfected by submerging in 70% ethanol for 5 min, washed with sterile distilled water, and placed in trays containing the sterile substrate Sunshine^®^-mix 3 plus perlite in a 3:1 ratio for germination. After germination, once seedlings reached 10–15 cm, they were placed in the greenhouse and inoculated in the roots with TiO_2_-NPs, TiO_2_-Bt, TiO_2_-Th, and TiO_2_-NPs-Bt/Th solutions at concentrations of 50, 100, and 150 μg/mL and the same concentrations of 10^6^ UFC/mL of Bt and 10^6^ spores/mL of Th, respectively. The experiment was carried out using twenty seedlings per treatment. The average temperature and relative humidity were 25–35 °C and 40–70%, respectively. After 15 days, a second application was performed, resulting in a total incubation period of 60 days.

#### 2.6.1. Evaluation of Physiological Parameters

Following the incubation period, root and shoot growth were measured using a methodology adapted from Kong et al. [25]. A Vernier caliper was used to measure shoot and root lengths from their intersection to their tips. After measuring the physiological parameters, twelve randomly selected seedlings were frozen in liquid nitrogen and stored at −80 °C for biochemical analysis. FW and DW were also measured following a standard procedure [26] with slight adaptations for each treatment and control. Eight randomly selected fresh seedlings per treatment were collected and weighed using an analytical balance for fresh weight determination. The seedlings were then covered with aluminum and placed in an oven at 105 °C for two hours and at 80 °C until a constant dry weight was obtained.

#### 2.6.2. Analysis of Antioxidant Responses

Plant tissue from six seedlings was frozen using liquid nitrogen and pulverized with a mortar and pestle to analyze the antioxidant responses of the chili pepper cultivars. A crude extract was prepared using 30 mg of pulverized plant tissue suspended in 5 mL of 50 mM sodium phosphate buffer, centrifuged, and used as an enzyme extract. Stress-related protein β-1,3-G and POX enzyme activities were evaluated following the procedure proposed by Barreto et al. [27]. β-1,3-G activity was determined by a colorimetric assay using a microplate reader (Agilent Biotek Synergy/HTX Multimode Reader, Santa Clara, CA, USA). The method was based on spectrophotometric detection of reducing sugars released by enzymatic hydrolysis, measured at 515 nm, following their reaction with DNS (3,5-Dinitrosalicylic acid). Quantification was performed using a calibration curve with glucose (0–200 µg/mL), and the activity was reported in nkats per gram of fresh weight (nkat/g FW). One nkat is defined as one nmol of D-glucose released from laminarin per second under the test conditions. POX quantification was performed in a 96-well microplate using 10 µL of plant extract mixed with 20 mM guaiacol, and the reaction was activated with 60 mM H_2_O_2_ at room temperature in a final volume of 250 µL. The absorbance was measured at 480 nm using a microplate reader. The variation in one absorbance unit per minute due to guaiacol oxidation was defined as one unit of peroxidase activity (1 UA) and was expressed per gram of fresh weight (UA/g FW). TPC was measured using the Folin–Ciocalteu method [28]. The sample (composed of the entire plant) was pulverized using liquid nitrogen and was used for phenolic compound extraction. For extraction, 35 mg of the sample was homogenized with extraction solvent (MeOH/water/formic acid 25:24:3 *v*:*v*:*v*), and the resultant mixture was sonicated (80% amplitude, 1 min), shaken (200 rpm, 20 min), and centrifuged (4000× *g*, 20 min). The reaction was measured at 750 nm using a microplate reader (Agilent Biotek Synergy/HTX multi-mode reader) and phenolic compounds were calculated based on a gallic acid standard curve (0–70 µg/mL).

#### 2.6.3. Assessment of Chlorophyll a, b, and Total Chlorophyll

Chlorophyll measurements were carried out based on the adapted procedures of Vatankhah et al. [29]. Leaf samples from the six remaining seedlings (0.1 g) were homogenized, mixed with 10 mL 80% acetone (*v*/*v*), centrifuged at 14,000× *g* for 15 min, and the supernatant was recovered. The supernatant of each sample was placed in a 96-well plate, and the absorbance was measured at 663 and 646 nm using a microplate reader (Agilent Biotek Synergy/HTX multi-mode reader). Chlorophyll a (Ca), chlorophyll b (Cb), and total chlorophyll (in μg per mL plant extract) were calculated using Equations (4) and (5), respectively [30]. Total chlorophyll content was calculated as the sum of chlorophyll a and b.(4)$$Ca=12.21A_{663}-2.81A_{646}$$(5)$$Cb=20.13A_{646}-5.03A_{663}$$

Determination of chlorophyll a content. Determination of chlorophyll-b content.

### 2.7. Statistical Analysis

Normality and homoscedasticity analyses were conducted, and the assumptions for performing an analysis of variance were verified. Analysis of variance was used to determine the effects of the nanofertilization and bio-nanofertilization processes. The analysis was performed using R software (R Core Team, Version: 4.5.0) and the RStudio integrated development environment (RStudio Team, Version: 2025.05.1+513). Data on bacterial and nanoparticle compatibility, physiological responses, TPC, enzyme activity, antioxidant activity, and compatibility were used to investigate the effects of TiO_2_, TiO_2_-Bt, TiO_2_-Th, and TiO_2_-Bt/Th. Tukey’s post hoc test was performed to identify specific pairwise differences between treatments.

## 3. Results

### 3.1. Characterization of TiO_2_-NPs

A schematic representation of the application of the synthesized TiO_2_ NPs is shown in Figure 1a. TiO_2_ was prepared using the molten salt method. This material was then used as a biofertilizer. Figure 1b shows the XRD patterns of TiO_2_-NPs. From these results, the (110), (101), (200), (111), (210), (211), (220), (002), (310), (221), (301), (112), (202), (212), (321), (400), (410), and (222) planes were indexed to the rutile crystallographic phase of TiO_2_ (JCPD #21-1276). According to Scherrer’s equation, the average crystallite size was calculated to be 41 ± 5 nm. No secondary phases or impurities were observed. Figure 1c illustrates the UV–Vis absorption spectra as a function of the wavelength range of 200–800 nm. The absorption spectra of TiO_2_-NPs exhibit a sharp edge at approximately 350 nm, corresponding to the intrinsic electron excitation from the valence band to the conduction band of TiO_2_. The Kubelka–Munk function was obtained from absorbance spectra [31]. The optical bandgap (Eg) was then calculated using Tauc’s equation [32]. The resulting Eg value 3.02 eV was similar to that reported previously [11]. The FT-IR spectrum of TiO_2_-NPs in the range 4000–480 cm^−1^ is shown in Figure 1d. In this plot, a band centered at 476 cm^−1^ corresponding to the Ti–O–Ti stretching vibration of the rutile phase can be observed [33].

The surface morphology of TiO_2_-NPs was analyzed using SEM. Figure 2a shows SEM images of the prepared material. The morphology consisted of nanocuboids of various sizes. Figure 2b shows the results of the specific surface area and pore volume distribution (inset) analyzed using the BET method. TiO_2_-NPs powders exhibit type IV N2 adsorption-desorption isotherms with a type H3 hysteresis loop [34]. The resulting BET-specific surface area (SBET) was 7.38 ± 5% m^2^/g, and the mean pore diameter was 8 ± 5% nm. Measurements of the z potential and particle size distribution in aqueous media of TiO_2_ are shown in Figure 2c and Figure 2d, respectively. The z-potential value was −50.06 ± 5% mV, representing good colloid stability [35]. The particle size distribution showed three dominant values (119, 515, and 5320 ± 5% nm), with a difference of one order of magnitude.

### 3.2. Compatibility of TiO_2_-NPs with PGPM

The compatibility of TiO_2_-NPs with Bs and Bt was analyzed based on their effects on kinetic growth (Appendix A) and CFUs formation (Appendix A). A significant increase in biomass production was observed (*p* < 0.05), with an increasing trend associated with increasing NP concentrations of 50, 100, and 150 μg/mL. The concentration of 150 µg/mL showed the most significant effect, reaching 2.74 ± 1.81 × 10^3^ CFU/mL for Bs and 2.99 ± 1.41 × 10^3^ CFU/mL for Bt. The compatibility of TiO_2_-NPs with Th was considered in spore production. A significant increase in spore production was recorded at 50 and 150 μg/mL, with values of 3.68 ± 2.77 × 10^3^ and 3.48 ± 1.67 × 10^3^ spores/mL, respectively, compared to the control (Appendix A). No significant differences were observed at 100 µg/mL.

### 3.3. Antioxidant Activity and In Vivo Toxicity of TiO_2_-NPs

The antioxidant activity of TiO_2_-NPs was evaluated using DPPH, ABTS, and H_2_O_2_ radicals. As shown in Figure 3a, treatment with 50, 100, and 150 μg/mL TiO_2_-NPs resulted in the inhibition of 49.86 ± 0.05, 55.89 ± 0.02, and 58.24 ± 0.15% DPPH radicals, respectively. Moreover, treatment with 50, 100, and 150 μg/mL TiO_2_-NPs resulted in the scavenging of 58.47 ± 0.20, 61.22 ± 0.10, and 62.50 ± 0.04% free radicals, respectively. Treatment with Qu was appraised as the positive control, and it was noted that treatment with 50 and 100 μg/mL inhibited the formation of 96.43 ± 0.35 and 97.15 ± 0.43%, respectively. In contrast, at 150 and 200 μg/mL, it scavenged 98.67 ± 0.20 and 99.66 ± 0.23% radicals, respectively. The same figure shows that treatment with 250 and 300 μg/mL of Qu resulted in complete inhibition of DPPH radicals.

As shown in Figure 3b, treatment with 50 and 100 μg/mL TiO_2_-NPs resulted in the scavenging of 48.82 ± 0.07 and 58.96 ± 0.11% ABTS radicals, respectively. Similarly, treatment with 150 and 200 μg/mL of TiO_2_-NPs occurred in the inhibition of 59.47 ± 0.04 and 59.64 ± 0.20% radicals, whereas at 250 and 300 μg/mL, treatment with TiO_2_-NPs inhibited the formation of 59.91 ± 0.25 and 60.57 ± 0.03% ABTS radicals. Again, Qu was appraised as the positive control, and it was determined that treatment with 50, 100, and 150 μg/mL scavenged 80.62 ± 1.64, 80.89 ± 0.11, and 81.39 ± 0.04% ABTS radicals, respectively. At higher concentrations, 200 and 250 μg/mL inhibited the generation of 81.57 ± 0.20 and 92.80 ± 0.25% ABTS radicals. Among the tested concentrations, treatment with 300 μg/mL of Qu exhibited the highest ABTS scavenging capacity, scavenging 93.46 ± 0.03% free radicals.

As observed in Figure 3c, treatment with TiO_2_-NPs at 50, 100, and 150 μg/mL resulted in the inhibition of 34.71 ± 0.33, 43.24 ± 0.09, and 47.39 ± 0.04% H_2_O_2_ radicals, respectively. In contrast to these results, treatment with 200 and 250 μg/mL resulted in scavenging of 97.11 ± 0.22 and 97.25 ± 0.17% H_2_O_2_ radicals, respectively. Similar antioxidant activity was noted during treatment with 300 μg/mL of TiO_2_-NPs, as indicated by the inhibition of 97.31 ± 0.31% H_2_O_2_ radicals. When evaluated using Qu, it was determined that treatment at 50 and 100 μg/mL scavenged 94.67 and 94.85% radicals, whereas at 150, 200, and 250 μg/mL, it scavenged 94.98 ± 0.01, 96.57 ± 0.08, and 96.68 ± 0.03% H_2_O_2_ radicals, respectively. Similarly, treatment with 300 μg/mL inhibited the generation of 97.34 ± 0.12% free radicals. The IC50 values of TiO_2_-NPs for DPPH, ABTS, and H_2_O_2_ radicals are presented in Table 1.

As noted in Appendix A, treatment with 50–300 μg/mL TiO_2_-NPs did not decrease the viability of *A. salina* nauplii. The Appendix A (Appendix A) shows the representative images of their biocompatibility. These results are opposite to those of treatment with K_2_Cr_2_O_7_, which decreased the viability of *A. salina* by 100% at the tested concentrations.

### 3.4. Effect of TiO_2_-NPs on Physiological Parameters of Chili Peppers

Key physiological variables were evaluated to determine the plant-stimulating effects of TiO_2_-NPs in combination with Bt and Th. The possible mechanisms involved in these effects are illustrated in Figure 4a. As shown in Figure 4b, plant height showed significant differences depending on the dose and combination with microorganisms. The greatest height was observed with 150 μg/mL TiO_2_-NPs (20 ± 2.4 cm), followed by treatment with 50 μg/mL plus Bt (18.1 ± 2.43 cm). In Figure 4c, it can be observed that root elongation was also favored by the high dose of TiO_2_-NPs (150 μg/mL), with the highest value recorded in combination with Th (7.05 ± 2.13 cm). As depicted in Figure 4d, a significant increase in leaf number was observed in the treatments that included NPs and microorganisms (*p* < 0.05), compared to the control and individual treatments. The highest number of leaves was obtained with 50 μg/mL of TiO_2_-NPs plus Bt (13.87 ± 3.37 leaves), a value comparable to that obtained during treatment with 150 μg/mL of TiO_2_-NPs (see Figure 4d). The FW and DW of the aboveground biomass increased significantly with the high dose of TiO_2_-NPs, showing similar effects in the combined treatments with microorganisms (see Figure 4e,f). This suggested a joint positive effect on biomass accumulation.

### 3.5. Biochemical Parameters

The enzymatic activities of β-1,3-G and POX were quantified as indicators of the activation of plant metabolism in response to stress. As represented in Figure 5b, a significant increase in β-1,3-G activity was observed in the TiO_2_-NPs treatments, especially when combined with Bt (3.72 ± 0.25 nkat/mg FW) and Th (2.85 ± 0.22 nkat/mg FW) (*p* < 0.05), compared to the control and individual microbial treatments. Interestingly, the synergistic effect generated enhanced antioxidant activity associated with the peroxidase enzyme, as shown in Figure 5c. POX activity also increased significantly, with the highest values in treatments combined with Bt (106.38 ± 7.14 UA/mg FW) and Th (81.42 ± 6.47 UA/mg FW). In contrast, the TPC decreased in most treatments compared to the control, especially in those that included only microorganisms (Figure 5d). Only treatments with 150 μg/mL TiO_2_-NPs plus Bt and 50 μg/mL TiO_2_-NPs showed levels similar to the control (see Figure 5d). All treatments resulted in significant increases in chlorophyll a, b, and total chlorophyll levels (*p* < 0.05), indicating improved photosynthetic capacity (see Figure 5d). Treatment with the microbial consortium without NPs presented the highest concentration of total chlorophyll (264.6 ± 49.1 mg/mg FW), followed by the combination of 150 μg/mL of TiO_2_-NPs with the consortium (210.8 ± 11.4 mg/mg FW) (see Figure 6).

## 4. Discussion

Precision agriculture has evolved as an efficient approach to agricultural management, enhancing crop productivity and sustainability through modern techniques such as nanotechnology. In precision agriculture, nanotechnology is considered for the development of nanofertilizers, pesticide delivery systems, plant sensors, and nanocatalysts. TiO_2_ is a white and opaque solid with high commercialization potential owing to its chemical stability, non-toxicity, and photocatalytic activity. Despite its wide use, the application of TiO_2_ in precision agriculture integrated with nanotechnology depends on the synthesis route and physicochemical features of the resultant nanomaterials.

The molten salt technique is a bottom-up approach, in which a mixture of salts becomes molten. In contrast to other methods, NPs have high purity, controlled size and morphology, and enhanced stability due to controlled experimental conditions. For nanobiotechnological applications, the size of NPs is a significant feature that dictates their capacity to penetrate plant tissues and cell membranes, where NPs lower than 100 nm are easily taken up by roots and leaves and are mobile in soil. The surface charge or zeta potential of NPs is another significant feature that determines the interaction of NPs with plant cell components or soil particles. During these phenomena, positively charged NPs (+30 mV) present a higher affinity for phospholipids or proteins, upregulate their uptake via endocytosis, and have a prominent capability to enhance plant growth under stress conditions. When negatively charged (−30 mV), NPs can interact with the positively charged components (e.g., pectin or lignin) of the plant cell wall together with basic amino acids (e.g., arginine or lysine). The specific surface area of NPs refers to the total area of their surfaces, where small-sized NPs have a higher surface-area-to-volume ratio than larger NPs. Together with other parameters, this feature is related to the reactivity, adsorption capacity, stability, and thermal properties of NPs. For precision agriculture applications, NPs with high surface areas are prone to enhance nutrient uptake, exhibit improved interactions with plant cell membranes, and upregulate protective biochemical pathways.

In this work, TiO_2_-NPs with negative surface charge (−50.06 ± 5% mV), heterogeneous size distribution, nanocuboid-type structure, 7.38 ± 5% m^2^/g surface area, and mean pore diameters of 8 ± 5% nm were synthesized. Together with these features, it was also determined that the average crystallite size and Eg of the synthesized TiO_2_-NPs were 41 ± 5% nm and 3.02 eV, respectively. Similar to the specific surface area, the crystallite size of the NPs is related to their surface interactions, catalytic activity, and reactivity. In contrast, determining the Eg of NPs intended for precision agricultural applications is necessary to understand their possible reaction to environmental stimuli, their capability to enhance photosynthesis, and to improve beneficial microbial activity and nutrient availability. The characterization results of the synthesized TiO_2_-NPs are challenging to compare with other studies in which such NPs have been applied for crop growth promotion, as they have partially addressed the complete determination of the physical, chemical, and optical features of the developed nanostructure.

PGPMs have garnered special attention because of their capacity to improve nutrient availability, physiological development, infection suppression, and stress tolerance. *Bacillus* and *Trichoderma* have unique features and mechanisms of action for crop growth and resilience. Contrary to other PGPMs, species from the genera *Bacillus* and *Trichoderma* exhibited enhanced capabilities for nitrogen fixation, phosphate solubilization, and nutrient absorption (see Figure 4a). In this study, the compatibility of TiO_2_-NPs with Bt or Th was analyzed before evaluating their effect on chili peppers. It was initially noted that treatment with TiO_2_-NPs increased the biomass of Bt (2.74 ± 1.81 × 10^3^ CFU/mL). Similarly, TiO_2_-NPs (3.68–3.48 × 10^3^ spores/mL) induced the spore production of Th. These results are challenging to compare with other studies since this is the first time that the compatibility of TiO_2_-NPs is analyzed with Bt and Th as PGPMs. Only one study reported that TiO_2_-NPs synthesized via the sol–gel route can inhibit the growth of PGPMs (i.e., *Pseudomonas aeruginosa*, *Klebsiella pneumoniae*, and *Serratia marcescens*) at 500–1000 μg/mL and are associated with their predominant activity against Gram-negative strains [36]. To date, no study has evaluated the compatibility of TiO_2_-NPs with Th. However, the positive effect of treatment with TiO_2_-NPs recorded in this study can be associated with results from other studies, where it has been demonstrated that green-synthesized oxide-based NPs (i.e., α-Fe_2_O_3_) can improve the growth of Th by upregulating the formation of mycelia, conidia, and conidiophores, together with the activity of chitinase and cellulase [37].

In precision agriculture, NPs are used to improve physiological parameters and biochemical responses. Here, it was noted that treatment with TiO_2_-NPs in synergy with Bt and Th improved root length (7.05 ± 2.13 cm), FW (5.90%), and DW (4.90%). The positive effect of treatment with TiO_2_-NPs and Bt and Th on the number of leaves is of great significance since it can occur in plants with enhanced photosynthetic capacity due to greater surface area for light absorption and improved nutrient uptake and utilization. Similarly, the beneficial effects of the treatment on root length and plant height can be associated with crops with improved water and nutrient acquisition, beneficial microbial interactions, and biomass production. Considering the latter, the recorded FW values can be used as an indicator of the positive effect of treatment with TiO_2_-NPs and Bt and Th on the growth and development of the cultivated chili peppers, indicating their vigor and improved capacity for photosynthesis, nutrient uptake, and metabolism. The determined DW values can also be considered an indicator of the nutritional content and the necessity of conducting other experiments to assess the long-term effects of the implemented treatments on the productivity and yield of the reported cultivars. The recorded effects of TiO_2_-NPs in synergy with Bt and Th are challenging to compare with other studies since this is the first time that the variabilities of FW and DW after treatment with these samples are reported, specifically in chili peppers with PGPMs.

Antioxidant responses include a series of physiological and biochemical events that diminish the possible detrimental effects of oxidative stress. Frequent antioxidant responses include the production of antioxidant compounds and enhancement of antioxidant enzyme activities. At the experimental level, the antioxidant capacity of NPs can also be evaluated through colorimetric assays, such as the DPPH and ABTS methods, supported by the implementation of free radicals as representatives of physiological oxidative stress processes. Here, it was observed that TiO_2_-NPs exerted moderate activity against DPPH, ABTS, and H_2_O_2_, with IC_50_ values of 5.39, 48.66, and 109.44 μg/mL. The scavenging capacity of TiO_2_-NPs for DPPH and ABTS has been widely reported. This correlates with their ability to donate electrons from their surfaces to neutralize the production of additional free radicals. However, the exact mechanism against H_2_O_2_ radicals remains unexplained and elusive; however, it has been noted that for oxide-based nanomaterials, H_2_O_2_ is transformed into water and oxygen during treatment, resulting in the mitigation of oxidative stress [38]. The calculated IC_50_ values from this study are challenging to compare with other studies because they have predominantly been reported for TiO_2_-NPs green synthesized with *Aloe vera* (0–50 ppm) [39], and *Tulbhagia violacea* (33.27–75.88 mg/mL) extracts [40], only considering the DPPH assay.

Nanomaterials exhibit intrinsic toxicity owing to their size, shape, composition, and surface chemistry. The interaction of nanoparticles (NPs) with biological structures can lead to the overgeneration of reactive oxygen species (ROS), DNA damage, or protein aberrations. In crops, NPs’ toxicity is initiated by their bioaccumulation in the roots, stems, or leaves, which subsequently impairs photosynthesis, nutrient uptake, enzymatic activity, growth, and development. Depending on their application route, NPs can also affect soil quality, and when transported into water systems through runoff, leaching, or soil erosion, they may jeopardize the overall health of aquatic organisms. The toxic effects of TiO_2_-NPs in crops remain unknown, as current scientific evidence has focused on their potential hazards to human health. Given our research group’s emphasis on nanotoxicology, a toxicity assay was conducted to assess the potential harmful effects of synthesized TiO_2_-NPs on *A. salina* nauplii. This initial step emphasizes the necessity of further research into the toxicity of NPs in crops, highlighting the need to explore their broader impacts on biological systems. It was observed that TiO_2_-NPs did not affect the anatomy or reduce the survival rate of *A. salina* nauplii. Current scientific evidence suggests using TiO_2_-NPs to prevent toxicological effects among *A. salina* nauplii. upon exposure to potentially toxic elements, rather than focusing on their potential detrimental effects. Despite these facts, the findings reported in this work are consistent with other studies, where it has been noted that TiO_2_-NPs (commercially obtained) did not affect the survival rate of *A. salina* at 1 and 5 mg L^−1^, respectively [41]. Nevertheless, possible differences with other studies can be attributed to differences in the synthesis method employed, physicochemical features, and tested concentrations of TiO_2_-NPs.

Phenols represent a wide range of secondary metabolites that are produced by plants. They defend themselves against phytopathogens and herbivores, regulate plant growth and pigmentation, and adapt to stress. This study reported that TiO_2_-NPs at 50 and 100 μg/mL caused varying TPC levels. In synergy with Bt and Th, it was determined that, at the same concentrations, treatment with TiO_2_-NPs caused 29.49 and 29.24 GA/g FW, respectively. TPC in chili pepper cultivars upon exposure to TiO_2_-NPs and PGPMs has not been reported; however, it has been reported that chili peppers infected with TMV and treated with TiO_2_-NPs exhibited a 13.5 times increase in TPC (mg GAE/g) [42]. β-1,3-G and POX are enzymes involved in stress responses during stress conditions, and the regulation of growth signals is required for plant growth and development. Here, TiO_2_-NPs were observed to occur in synergy with Bt and Th, resulting in treatment at 3.72 ± 0.25 and 2.85 ± 0.22 nkat/mg FW levels, respectively. Evidence regarding the application of TiO_2_-NPs in chili peppers is limited. Similar results were observed when chili peppers were treated with TiO_2_-NPs and Bt and Th, exhibiting 106.38 ± 7.14 and 81.42 ± 6.47 AU/mg FW levels. Again, only one study has reported that treatment with TiO_2_-NPs synthesized via the sol–gel route exerted biostimulant properties during induced infections with PHYVV and TMV strains by enhancing 10.7 times the activity of POX, together with other enzymes [42]. The use of TiO_2_-NPs and Bt and Th also occurred in enhanced levels of total chlorophyll (210.8 ± 11.4 mg/mg FW), which is of wide importance since it poses a significant role in photosynthetic efficiency, stress tolerance, and crop resilience. It has been demonstrated that under controlled pot conditions, the co-application of silicon dioxide nanoparticles (SiO_2_) with PGPR rhizobacteria produced a synergistic effect, significantly enhancing wheat growth and yield as well as its drought tolerance. Increases in biomass, fresh weight, dry weight, and chlorophyll-a and chlorophyll-b contents were observed compared to controls. Furthermore, improvements were seen in relative water content (71.66%), gas exchange, nutrient absorption, and osmolyte synthesis. Concurrently, a significant overexpression of antioxidant enzymes such as superoxide dismutase (60.49%), peroxidase (55.99%), and catalase (81.69%) was recorded [43].

## 5. Conclusions

This study demonstrated the effect of TiO_2_-NPs on the physiological parameters and biochemical responses of chili peppers using *B. thuringiensis* and *T. harzianum* as PGPMs for the first time. Characterization techniques revealed that TiO_2_-NPs exhibit a nanocuboid-type structure, H3 hysteresis loops, 7.38 ± 5% m^2^/g surface area, and negative surface charge (−50.06 mV). Compatibility assays between TiO_2_-NPs and PGPMs demonstrated that treatment at 50, 100, and 150 μg/mL did not compromise the CFU or spore production of *B. thuringiensis* (1.56–2.92 × 10^8^ CFU/mL) and *T. harzianum* (2.50–3.90 × 10^8^ spores/mL), respectively. This work also reported that TiO_2_-NPs exerted moderate activity against DPPH radicals (IC_50_ 52.39 μg/mL), ABTS (IC_50_ 48.66 μg/mL), and H_2_O_2_ (IC_50_ 109.94 μg/mL) radicals. Against *A. salina* nauplii, treatment with TiO_2_-NPs did not decrease their viability within 50–300 μg/mL, suggesting their biocompatibility. When treated with 100 μg/mL TiO_2_-NPs and inoculated with *B. thuringiensis* and *T. harzianum*, chili peppers presented reduced leaf numbers (11.50) and plant height (15.72 cm), but improved root length (6.57 cm), FW (5.90%), and DW (4.90%). Contrary to these findings, treatment with 50 μg/mL TiO_2_-NPs and inoculation with *B. thuringiensis* and *T. harzianum* upregulated the activity of β-1,3-G (2.45 nkat/g FW) and POX (69.90 UA/g FW). The treatment also increased the TPC (29.50 GA/g FW). Chlorophyll a, b, and total chlorophyll were also influenced by TiO_2_-NP, Bt, and Th, occurring at 210.8 ± 11.4 mg/mg FW. The data retrieved from this study show, for the first time, the physiological and biochemical responses of chili pepper cultivars to exposure to TiO_2_-NPs and PGPMs. In addition, this study expands the scientific evidence regarding the synthesis, physicochemical characterization, and in vitro, in vivo, and in planta performance of non-metal-based nanofertilizers for precision agriculture purposes.

## Figures and Tables

**Figure 1 antioxidants-14-00707-f001:**
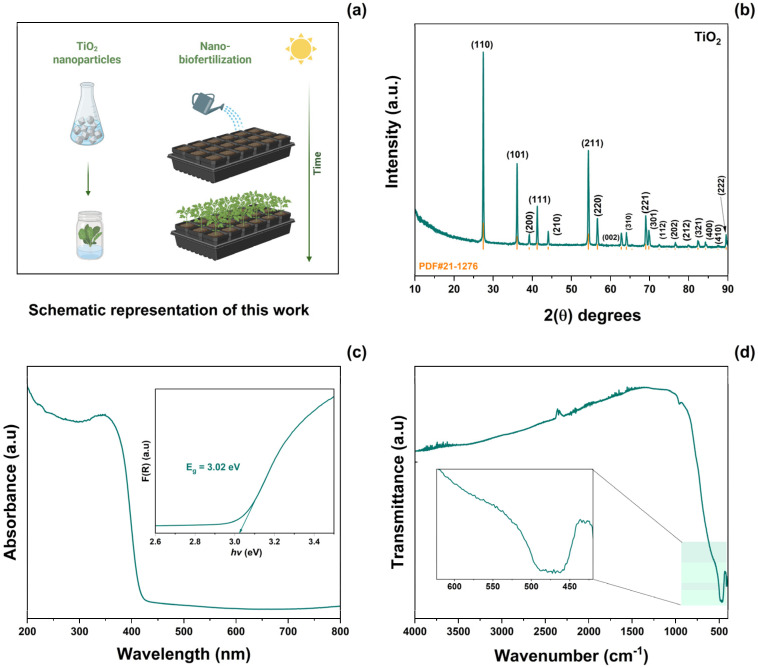
(**a**) Schematic representation of this work, (**b**) XRD pattern, (**c**) absorbance spectrum, and (**d**) FT-IR spectrum of as-prepared TiO_2_-NPs.

**Figure 2 antioxidants-14-00707-f002:**
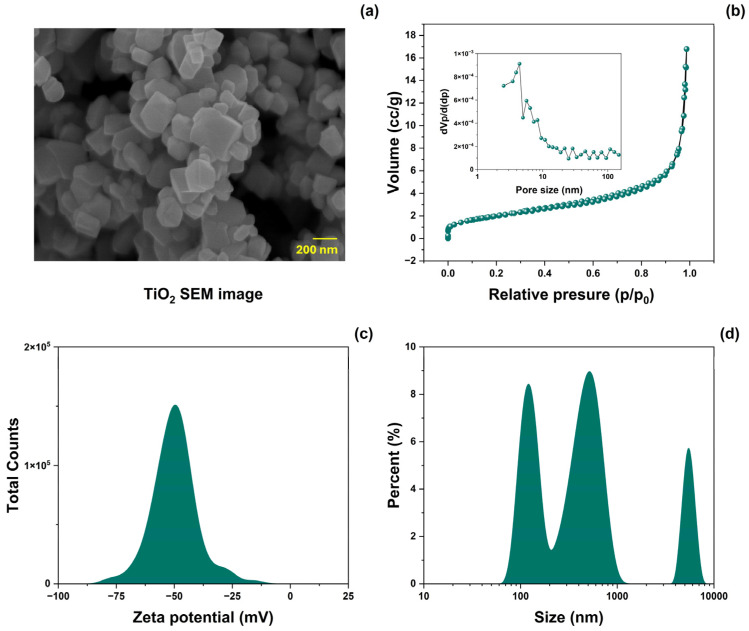
(**a**) SEM image, (**b**) N_2_-adsorption-desorption isotherm plots (inset: pore size distribution), (**c**) zeta potential (ζ), and (**d**) size distribution analysis of TiO_2_-NPs.

**Figure 3 antioxidants-14-00707-f003:**
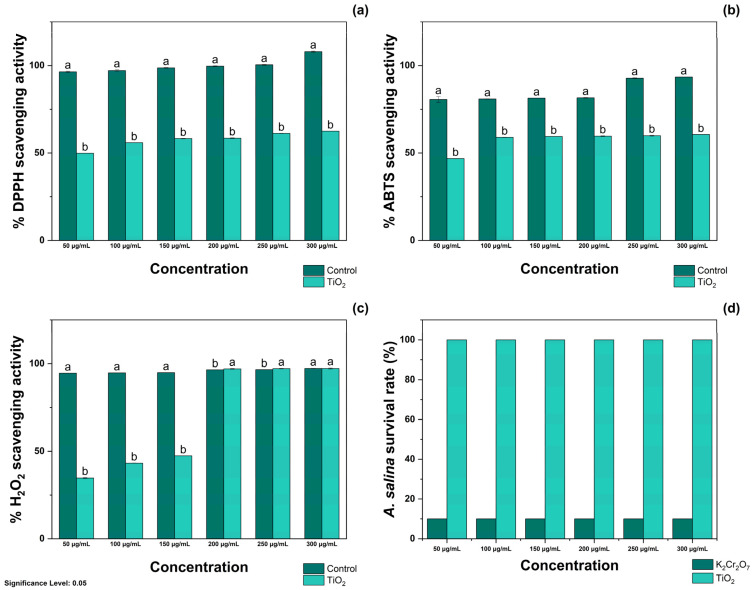
Antioxidant activity of TiO_2_-NPs against (**a**) DPPH, (**b**) ABTS, and (**c**) H_2_O_2_ radicals and their effects on (**d**) the survival rate of *A. salina*. Significance Level 0.05: Confidence level based on a 5% error considered in statistical evaluations.

**Figure 4 antioxidants-14-00707-f004:**
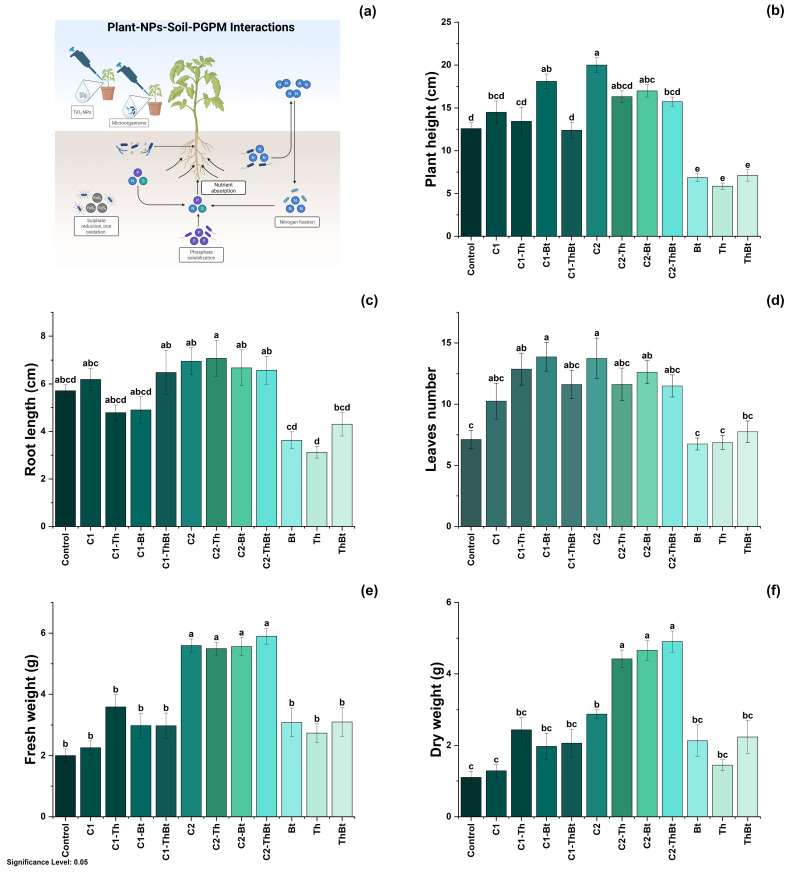
Physiological response to synergy between TiO_2_-NPs and PGPMs. (**a**) Representative diagram of the induction of the physiological process and effect of TiO_2_-NPs /PGPMs on (**b**) plant height, (**c**) root length, (**d**) leaf number, (**e**) FW, and (**f**) DW of chili pepper cultivars. The data are represented as the mean ± standard deviation (SD) from six independent experiments, with the resulting Tukey’s test group analysis showing a *p*-value of less than 0.05. Significance Level 0.05: Confidence level based on a 5% error considered in statistical evaluations.

**Figure 5 antioxidants-14-00707-f005:**
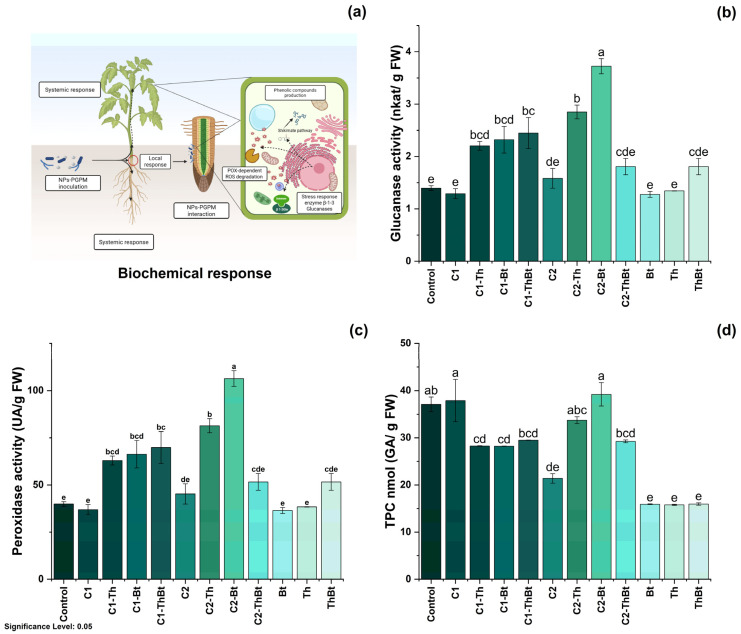
Biochemical response of synergy between TiO_2_-NPs and PGPMs. (**a**) Schematic representation of the biochemical response and effect of TiO_2_-NPs/PGPMs on (**b**) β-1,3-glucanase enzyme activity, (**c**) peroxidase activity, and (**d**) TPC production. The data are represented as the mean ± standard deviation (SD) from six independent experiments, with the resulting Tukey’s test group analysis showing a *p*-value of less than 0.05. Significance Level 0.05: Confidence level based on a 5% error considered in statistical evaluations.

**Figure 6 antioxidants-14-00707-f006:**
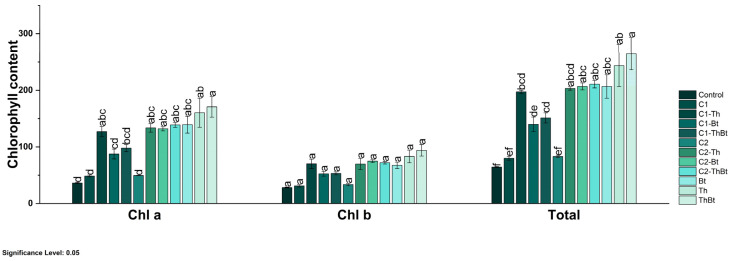
Chlorophyll content in chili peppers treated with TiO_2_-NPs and PGPMs. The data are represented as the mean ± standard deviation (SD) from six independent experiments, with the resulting Tukey’s test group analysis showing a *p*-value of less than 0.05. Significance Level 0.05: Confidence level based on a 5% error considered in statistical evaluations.

**Table 1 antioxidants-14-00707-t001:** IC_50_ values of TiO_2_-NPs and Qu against DPPH, ABTS, and H_2_O_2_ radicals. Concentrations are expressed in μg/mL.

Sample	DPPH	ABTS	H_2_O_2_
TiO_2_-NPs	52.39	48.66	109.94
Qu	3.57	3.04	2.64

## Data Availability

Data generated from this study can be provided on consultation with the authors upon reasonable request.

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
