# Peer review of "Synergistic Effect of TiO2-Nanoparticles and Plant Growth-Promoting Microorganisms on the Physiological Parameters and Antioxidant Responses of Capsicum annum Cultivars"

_antioxidants, 2025, doi:10.3390/antiox14060707_

Round 1
Reviewer 1 Report
The overall quality of this manuscript is not too bad, and shows some positive results toward future readers
Main comements
1) The intrinisic toxicity of the Nanoparticles should be explained as well to show the background value
2) The performance comparison in Fig 3, Fig 4 and Fig 5 should be cross-cited and compared with other similar studies in this field.
3) The results in section 3.6 is not so convinced and may cause many doubts, deep discussion is required to make it clear and reasonable as well.
Author Response
Comment 1:
The overall quality of this manuscript is not too bad, and shows some positive results toward future readers.
Answer: Thank you for your time and for providing comments.
Comment 2:
1) The intrinisic toxicity of the Nanoparticles should be explained as well to show the background value
Answer: Thank you to the reviewer for your comment. This information has been included in the revised version of the discussion of the manuscript.
Comment 3:
2) The performance comparison in Fig 3, Fig 4 and Fig 5 should be cross-cited and compared with other similar studies in this field.
Answer: Thank you for your careful observation. We have conducted an extensive review of the available scientific literature, finding that the evidence regarding the use of TiO2-NPs in promoting the physiological and biochemical responses of chili pepper cultivars is scarce. This has been mentioned when appropriate, emphasizing that TiO2-NPs have been considered for other crops but rarely for chili peppers. Regarding the antioxidant activity, it was also mentioned in the original version that the calculated scavenging performance is challenging to compare, considering the calculated IC50, which has been reported only for green-synthesized TiO2-NPs and for the DPPH assay. This fact is important to consider, since we performed three antioxidant assays (DPPH, ABTS, and H2O2), and as mentioned at the beginning of the discussion section, because the effect of NPs in crops varies according to the synthesis route and the determined physical and chemical parameters.
Comment 4:
3) The results in section 3.6 is not so convinced and may cause many doubts, deep discussion is required to make it clear and reasonable as well.
Answer: Thank you for pointing out this information. We have included information related to the synergistic interaction between SiO2 nanoparticles and growth-promoting rhizobacteria, which demonstrates its effect on physiological development, the increase in photosynthetic pigments, and enzymatic activities in response to stress.
Reviewer 2 Report
This manuscript explores a timely and important topic in the field of sustainable agriculture. The combined use of TiO₂ nanoparticles with plant growth-promoting microorganisms (PGPMs), specifically Bacillus thuringiensis and Trichoderma harzianum, to enhance physiological and antioxidant responses in Capsicum annuum cultivars presents a promising approach. The integration of nanotechnology with microbial biofertilization is innovative, and the experimental design appears solid. However, some sections of the manuscript would benefit from additional clarification and detail to improve overall clarity and scientific rigor.
Keywords: The current keywords (antioxidant response, Capsicum annuum, plant growth-promoting microorganisms) are already present in the title. To improve the discoverability and impact, include additional keywords that are not part of the title.
Methods
The description of the statistical analysis is brief and lacks information on whether key assumptions (e.g., normality, homoscedasticity) were verified prior to conducting ANOVA and Tukey’s post hoc test. Please clarify whether these assumptions were tested and indicate the significance level used (e.g., p < 0.05).
Although the manuscript states that 20 seedlings were used per treatment, it is unclear how many samples were used for each specific measurement (e.g., enzymatic assays, chlorophyll quantification). Please specify the number of biological and technical replicates for each type of analysis.
Results
Some results (e.g., CFU counts, spore production, chlorophyll content) show high standard deviations. It would be helpful for the authors to comment on this variability, is it biologically expected, or might it suggest technical inconsistencies?
The term “synergistic effects” is used frequently; however, these claims are difficult to verify in the absence of statistical tests for interaction (e.g., two-way ANOVA). Did the authors perform such analyses to support the claim of synergy between TiO₂-NPs and PGPMs?
Discussion
The discussion primarily highlights positive results, while negative or ambiguous findings are not addressed. Were there any instances in which TiO₂-NPs had neutral or inhibitory effects, especially at higher concentrations?
Figures
The figures are generally clear, well-labeled, and informative. The SEM and spectroscopy images (Figures 1–2) are of high quality.
For Figures 3–6, please include indicators of statistical significance (p < 0.05) and specify the statistical tests used in the figure legends.
Supplementary Figure S3 could be cited into the main manuscript.
Author Response
Dear reviewer, thank you very much for taking the time to review this manuscript. Please find the detailed responses below and the corresponding revisions/corrections highlighted/in track changes in the re-submitted files.
Comment 1:
Keywords: The current keywords (antioxidant response, Capsicum annuum, plant growth-promoting microorganisms) are already present in the title. To improve the discoverability and impact, include additional keywords that are not part of the title.
Answer: Thank you to the reviewer for their comment. The new keywords are: Biocompatibility, Total phenolic compounds, Titanium dioxide, Antioxidant enzymes, physiological response.
Comment 2:
The description of the statistical analysis is brief and lacks information on whether key assumptions (e.g., normality, homoscedasticity) were verified prior to conducting ANOVA and Tukey’s post hoc test. Please clarify whether these assumptions were tested and indicate the significance level used (e.g., p < 0.05).
Answer: Thank you to the reviewer for their comment. Before conducting the analysis of variance, the assumptions of normality and homoscedasticity were verified. Data normality was assessed using the Shapiro-Wilk test, which produced p-values < 0.05 for all analyzed variables, indicating a significant deviation from normal distribution. Homoscedasticity was evaluated using the Breusch-Pagan test, yielding p > 0.05, suggesting that the assumption of equal variances among groups was met.
Comment 3:
Although the manuscript states that 20 seedlings were used per treatment, it is unclear how many samples were used for each specific measurement (e.g., enzymatic assays, chlorophyll quantification). Please specify the number of biological and technical replicates for each type of analysis.
Answer: Thank you to the reviewer for their comment. We have included the distribution of plants by treatment in both the methodology and the results descriptions.
Comment 4:
Some results (e.g., CFU counts, spore production, and chlorophyll content) show high standard deviations. The authors should comment on this variability. Is it biologically expected, or might it suggest technical inconsistencies?
Answer: Thank you to the reviewer for their comment. The variability observed in experiments involving biological organisms is a statistically expected phenomenon. This variation primarily depends on the specific physiological conditions of each microbial culture (bacterial or fungal) and the intrinsic genetic variability of each plant, including hybrids. The experiment was conducted using hybrid seeds, and although they are genetically similar, their metabolic responses vary. Despite the experimental controls, fluctuations like these are common in biological studies and should be regarded as a natural aspect of the behavior of living systems.
Comment 5:
The term “synergistic effects” is used frequently; however, these claims are difficult to verify in the absence of statistical tests for interaction (e.g., two-way ANOVA). Did the authors perform such analyses to support the claim of synergy between TiO₂-NPs and PGPMs?
Answer: Thank you to the reviewer for their comment. Certainly, understanding the mechanisms of synergy between nanoparticles and PGPM can be explored through gene expression profiles and their relation to the metabolic pathways associated with growth. In this study, the synergistic effect was linked to the statistical analysis generated by comparing the presence and absence of microorganisms in treatments that included nanoparticles with those that included only beneficial microorganisms.
Comment 6:
The discussion primarily highlights positive results, while negative or ambiguous findings are not addressed. Were there any instances in which TiO₂-NPs had neutral or inhibitory effects, especially at higher concentrations?
Answer: Thank you to the reviewer for their comment. Before the plant study, we assessed the potential toxic effects of nanoparticles on beneficial microorganisms, taking into account the reported adverse effects associated with other metal oxide-based nanoparticles. However, our results showed no toxicity within the concentration ranges used; rather, a positive effect on microbial biomass proliferation was apparent. When we expanded the evaluation to a plant model, this microbial stimulation was reflected in an improved physiological response in the treated plants.
Comment 7:
The figures are generally clear, well-labeled, and informative. The SEM and spectroscopy images (Figures 1–2) are of high quality.
Answer: Thank you to the reviewer for their comment; we appreciate it.
Comment 8:
For Figures 3–6, please include indicators of statistical significance (p < 0.05) and specify the statistical tests used in the figure legends.
Answer: Thank you to the reviewer for their comment. All the figures include the p-value; we are now increasing the font size.
Comment 9:
Supplementary Figure S3 could be cited into the main manuscript.
Answer: Thank you to the reviewer for their comment. Figure S3 is now cited in the main manuscript.